# Single spin localization and manipulation in graphene open-shell nanostructures

Jingcheng Li[1], Sofia Sanz[2], Martina Corso[2,3], Deung Jang Choi[2,3,4], Diego Peña[5], Thomas Frederiksen[2,4] & Jose Ignacio Pascual[1,4]

Turning graphene magnetic is a promising challenge to make it an active material for spintronics. Predictions state that graphene structures with specific shapes can spontaneously develop magnetism driven by Coulomb repulsion of $\pi$-electrons, but its experimental verification is demanding. Here, we report on the observation and manipulation of individual magnetic moments in graphene open-shell nanostructures on a gold surface. Using scanning tunneling spectroscopy, we detect the presence of single electron spins localized around certain zigzag sites of the carbon backbone via the Kondo effect. We find near-by spins coupled into a singlet ground state and quantify their exchange interaction via singlet-triplet inelastic electron excitations. Theoretical simulations picture how electron correlations result in spin-polarized radical states with the experimentally observed spatial distributions. Extra hydrogen atoms bound to radical sites quench their magnetic moment and switch the spin of the nanostructure in half-integer amounts. Our work demonstrates the intrinsic $\pi$-paramagnetism of graphene nanostructures.

[1] CIC nanoGUNE, 20018 Donostia-San Sebastián, Spain. [2] Donostia International Physics Center (DIPC), 20018 Donostia-San Sebastián, Spain. [3] Centro de Física de Materiales CFM/MPC (CSIC-UPV/EHU), 20018 Donostia-San Sebastián, Spain. [4] Ikerbasque, Basque Foundation for Science, 48013 Bilbao, Spain. [5] Centro Singular de Investigación en Química Biolóxica e Materiais Moleculares (CiQUS), and Departamento de Química Orgánica, Universidade de Santiago de Compostela, Santiago de Compostela 15782, Spain. Correspondence and requests for materials should be addressed to J.I.P. (email: ji.pascual@nanogune.eu)

Among the many applications predicted for graphene, its use as a source of magnetism is the most unexpected one, and an attractive challenge for its active role in spintronic devices[1]. Generally, magnetism is associated to a large degree of electron localization and strong spin–orbit interaction. Both premises are absent in graphene, a strongly diamagnetic material. The simplest method to induce magnetism in graphene is to create an imbalance in the number of carbon atoms in each of the two sublattices, what, according to the Lieb's theorem for bipartite lattices[2], causes a spin imbalance in the system. This can be done by either inserting defects that remove a single $p_z$ orbital[3–6] or by shaping graphene with zigzag (ZZ) edges[7,8]. However, magnetism can also emerge in graphene nanostructures where Lieb's theorem does not apply[9,10]. For example, in $\pi$-conjugated systems with small band gaps, Coulomb repulsion between valence electrons forces the electronic system to reorganize into open-shell configurations[11] with unpaired electrons (radicals) localized at different atomic sites. Although the net magnetization of the nanostructures may be zero, each radical state hosts a magnetic moment of size $\mu_B$, the Bohr magneton, turning the graphene nanostructure paramagnetic. This basic principle predicts, for example, the emergence of edge magnetization originating from zero-energy modes in sufficiently wide ZZ[12–14] and chiral[15,16] graphene nanoribbons (chGNRs).

The experimental observation of spontaneous magnetization driven by electron correlations is still challenging, because, for example, atomic defects and metal impurities in the graphene structures[17–19] hide the weak paramagnetism of radical sites[20]. Scanning probe microscopies can spatially localize the source states of magnetism[6,19], but they require both atomic-scale resolution and spin-sensitive measurements. Here we achieve these conditions to demonstrate that atomically defined graphene nanostructures can host localized spins at specific sites and give

rise to the Kondo effect[21,22], a many-body phenomenon caused by the interaction between a localized spin and free conduction electrons in its proximity. Using a low-temperature scanning tunneling microscope (STM) we use this signal to map the spin localization within the nanostructure and to detect spin–spin interactions.

## Results

**Formation of GNR nanostructures**. The graphene nanostructures studied here are directly created on a Au(111) surface by cross-dehydrogenative coupling of adjacent chiral GNRs (chGNRs)[23]. We deposited the organic molecular precursors 2,2′-dibromo-9,9′-bianthracene (Fig. 1a) on a clean Au(111) surface, and annealed stepwisely to 250 °C (step 1 in Fig. 1a) to produce narrow (3,1)chGNRs, i.e. ribbons that alternate three zig-zag sites with one armchair along the edge[24]. They are semiconductors with a band-gap of 0.7 eV and show two enantiomeric forms on the surface[25]. By further annealing the substrate to 350 °C (step 2 in Fig. 1a), chGNRs fuse together into junctions, as shown in Fig. 1b. The chGNR junctions highlighted by dashed rectangles are the most frequently found in our experiments. They consist of two chGNRs with the same chirality linked together by their termination (Fig. 1c). The creation of this stable nanostructure implies the reorganization of the carbon atoms around the initial contact point[26] into the final structure shown in Fig. 1d, as described in Supplementary Note 1.

In Fig. 1b, certain regions of the junctions appear brighter when recorded at low sample bias, reflecting enhancements of the local density of states (LDOS) close to the Fermi level. Interestingly, the precise location of the bright regions is not unique, but can be localized over the pentagon cove (PC) site (Type 1, Fig. 1e), over the terminal ZZ site of the junction (Type

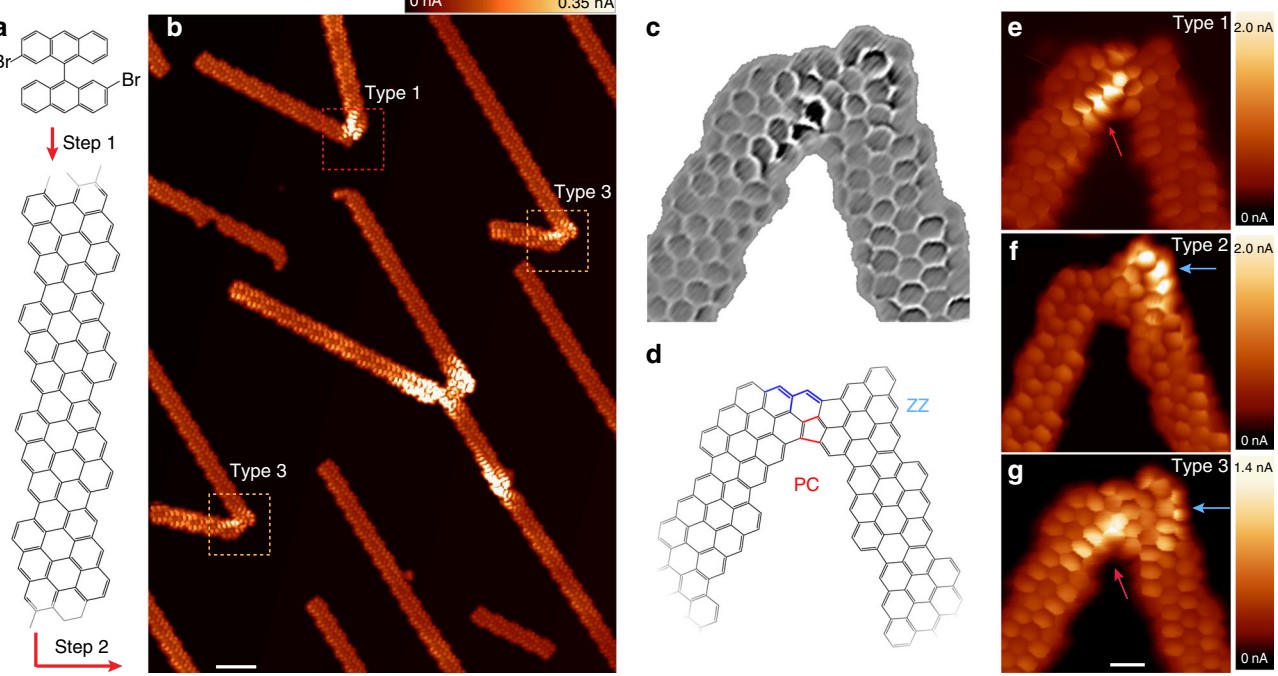

**Fig. 1** Formation of GNR junctions. **a** Model structures of the organic precursor 2,2′-dibromo-9,9′-bianthracene and of the on-surface synthesized (3,1) chGNR after Ullmann-like C–C coupling reaction and cyclodehydrogenation on Au(111). **b** Constant-height current images ($V = 2$ mV, scale bar: 2 nm) showing joint chGNR nanostructures, with an angle of ~50°, obtained after further annealing the sample. A CO-functionalized tip was used to resolve the chGNR ring structure. Dashed boxes indicate the most characteristic chGNR junctions, whose structure is shown in panels **c**, **d**. **c** Laplace-filtered image of the junction shown in panel **g** to enhance the backbone structure, and (**d**) model structure of the junction. PC labels the pentagonal cove site and the ZZ the zigzag site. **e–g** Constant-height current images ($V = 8$ mV, scale bar 0.5 nm) of the three types of chGNR junctions with same backbone structures but with different LDOS distribution

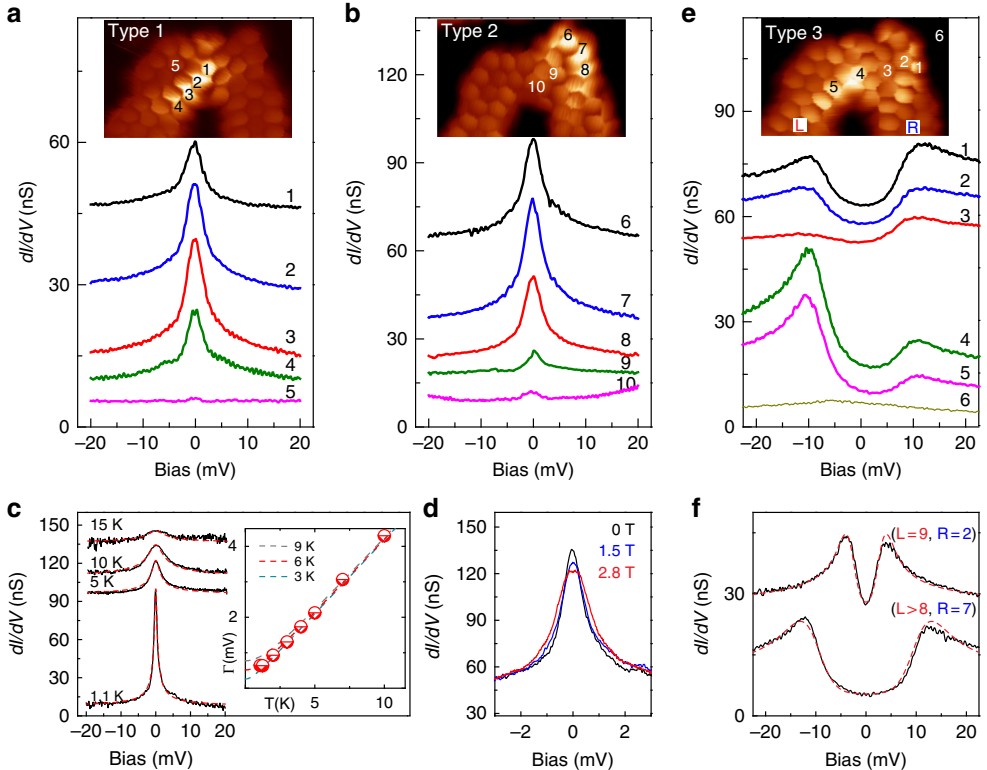

**Fig. 2** Zero bias spectral features. **a**, **b** Kondo resonances over the bright regions of Type 1 and Type 2 junctions, respectively. The zero-bias peaks are mostly detected over four PC rings of Type 1 junctions and over three ZZ rings of Type 2 junctions. **c** Temperature dependence of the Kondo resonance. All spectra were measured over the same PC site. The half width at half maximum (HWHM) at each temperature is extracted by fitting a Frota function (red dashed lines)[49] and corrected for the thermal broadening of the tip[28]. The temperature dependence of HWHM was simulated with the empirical expression $\frac{1}{2}\sqrt{(\alpha k_B T)^2 + (2k_B T_K)^2}$[29], which reproduces the experimental results with a Kondo temperature $T_K \sim 6\,K$ and $\alpha = 9.5$ (Supplementary Note 3). **d** Magnetic field dependence of a Kondo resonance (over the same PC site) at the field strengths indicated in the figure. **e** Double-peak features around zero bias over Type 3 junctions. **f** Split-peak d$I$/d$V$ features for nanostructures with different sizes, determined by the number of precursor units in each chGNR, labeled L and R in **e**. The gap width increases with the length of the ribbons (see Supplementary Fig. 16a in Supplementary Note 8). The red dashed lines are fits to our spectra using a model for two coupled spin-1/2 systems[31]. The spectra in **c**, **d** were acquired with a metal tip, while the others with a CO-terminated tip

2, Fig. 1f), or over both (Type 3, Fig. 1g). Supplementary Note 2 quantifies the probability of finding each type of junction. Despite these different LDOS distributions in the three types of junctions, they all have the same carbon lattice arrangement, shown in Fig. 1d. Such low-energy LDOS enhancements are absent over bare chGNRs segments due to their semiconductor character, and only the bare hexagonal backbone is resolved.

**Spectral features around zero-bias**. To understand the origin of the enhanced LDOS at the ZZ and PC sites, we recorded differential conductance spectra (d$I$/d$V$) on the three types of junctions. Spectra on the bright sites of Type 1 and 2 junctions show very pronounced zero-bias peaks (Fig. 2a, b) localized over the bright sites (spectra 1–4, and 6–8), and vanishing rapidly in neighbor rings (spectra 5, 9, and 10). These are generally ascribed as Abrikosov–Suhl resonances due to the Kondo effect, and named as Kondo resonances[21,22]. Their observation is a proof of a localized magnetic moment screened by conduction electrons[27,28] (see Supplementary Note 3). The relationship between the observed peaks and the Kondo effect is proven by measurements of d$I$/d$V$ spectra at different temperatures (Fig. 2c) and magnetic fields (Fig. 2d). The resonance line width increases with temperature following the characteristic behavior of a Kondo-screened state with a Kondo temperature $T_K \sim 6\,K$[28,29] (Fig. 2c), and broadens with magnetic field as expected for a spin-1/

2 system (Fig. 2d). Hence, the bright regions are caused by the localization of a single magnetic moment.

Junctions with two bright regions (Type 3) show different low-energy features: two peaked steps in d$I$/d$V$ spectra at $\sim\pm10$ meV (Fig. 2e). The steps appear always symmetric with respect to zero bias, and at the same energy over the terminal ZZ segment and over the PC region for a given nanostructure, while vanish quickly away from these sites. Based on the existence of localized spins on bright areas of Type 1 and 2 junctions, we attribute such bias-symmetric peaked steps to the excitation of two exchange coupled spins localized at each junction site by tunneling electrons. The exchange interaction tends to freeze their relative orientation, in this case antiferromagnetically into a singlet ground state. Electrons tunneling into the coupled spin system can inelastically excite a spin reversal in any of them when their energy equals the exchange coupling energy between the spins, i.e., $eV \geq J$. Usually, such singlet–triplet inelastic excitation is revealed in d$I$/d$V$ spectra as steps at the onset of spin excitations[30], from which one can directly determine the strength of the exchange coupling $J$ between the spins. Here, the inelastic spectra additionally show asymmetric peaks on top of the excitation onsets, with a pronounced logarithmic fall off for biases above. Such peaked steps are characteristic of Kondo-like fluctuations of the spin once the anisotropy energy has been overcome by tunneling electrons (i.e. out of equilibrium)[31–34]. The more pronounced signal for either particle tunneling (over

ZZ) or for hole tunneling (over PC) indicates the spins system lies out of particle-hole symmetry point, with $E_F$ closer to the corresponding singly unoccupied or singly occupied (SO) state, respectively. Hence, the gap between $dI/dV$ peaks in Fig. 2e is a measure of the interaction strength between the two localized spins.

Interestingly, the spectral gap in Type 3 junctions increases with the length of the connecting ribbons. In Fig. 2f we compare low-energy spectra of two junctions with different chGNR lengths. Although the atomic structures of both junctions are identical, the one with shorter ribbons (upper curve; 9 and 2 precursor units) displays a smaller gap than the junction of longer chGNRs (lower curve; >8 and 7 units). Fitting the spectra with a model of two coupled spin-1/2 systems[31], one obtains the exchange coupling $J = 2.7$ (9.9) meV for the upper (lower) spectrum.

**Theory simulations to uncover the origin of spin polarization**. To explain the emergence of localized spins, we simulated the spin-polarized electronic structure of chGNR junctions using both density functional theory (DFT) and mean-field Hubbard (MFH) models (see Supplementary Notes 6 and 7). Figure 3a shows the spin-polarization of a junction of Fig. 1d. The ground state exhibits a net spin localization at the ZZ and PC regions with opposite sign, which is absent in the bare ribbons. The spin distribution along the edge sites reproduces the distribution of $dI/dV$ signal measured in Type 3 junctions. This supports that this signal is an intrinsic effect of junction edge sites, rather than caused by defects or other atomic species.

The origin of the spontaneous magnetization can be rationalized by considering the effect of Coulomb correlations between

$\pi$-electrons as described within a tight-binding (TB) model (Fig. 3b). The spin distribution is related to the appearance of two junction states inside the gap of the (3,1)-chGNR electronic bands, localized at the PC and ZZ sites, respectively (Fig. 3c). These are split-off states from the VB of the (3,1)-chGNR, which lies close below $E_F$[25]. In the absence of electron–electron correlations, these two states conform the highest occupied (HO) and lowest unoccupied (LU) molecular states of the nanostructure. Due to the large degree of localization (Supplementary Figs. 10 and 11), the Coulomb repulsion energy $U_{HH}$ between two electrons in the HO state becomes comparable with the energy difference $\delta$ between the two localized levels. Hence, in a simplified picture, the two electrons find a lower-energy configuration by occupying each a different, spatially separated in-gap state. These two states become SO, spin-polarized (i.e., they have a net magnetic moment), and exchange coupled as schematically illustrated in Fig. 3b. Similar conclusions regarding the emergence of radical states at PC and ZZ sites can also be reached by applying the empirical Clar's aromatic $\pi$-sextet rule to the close-shell structure of Fig. 1d, as described in the Supplementary Note 4.

According to both DFT (Fig. 3a) and MFH (Supplementary Fig. 14) the magnetic moments are antiferromagnetically aligned into a singlet ground state. Therefore, the inelastic features in $dI/dV$ spectra found over Type 3 junctions (Fig. 2e) are associated to singlet–triplet excitations induced by tunneling electrons. In fact, the smaller excitation energy found for the smaller ribbons in both theory and experiment (Supplementary Note 8) agrees with a weaker exchange interaction due to a larger localization of the spin-polarized states. Alternative scenarios for peaks around $E_F$, such as single-particle states or Coulomb-split radical states[6], would show the opposite trend with the system size.

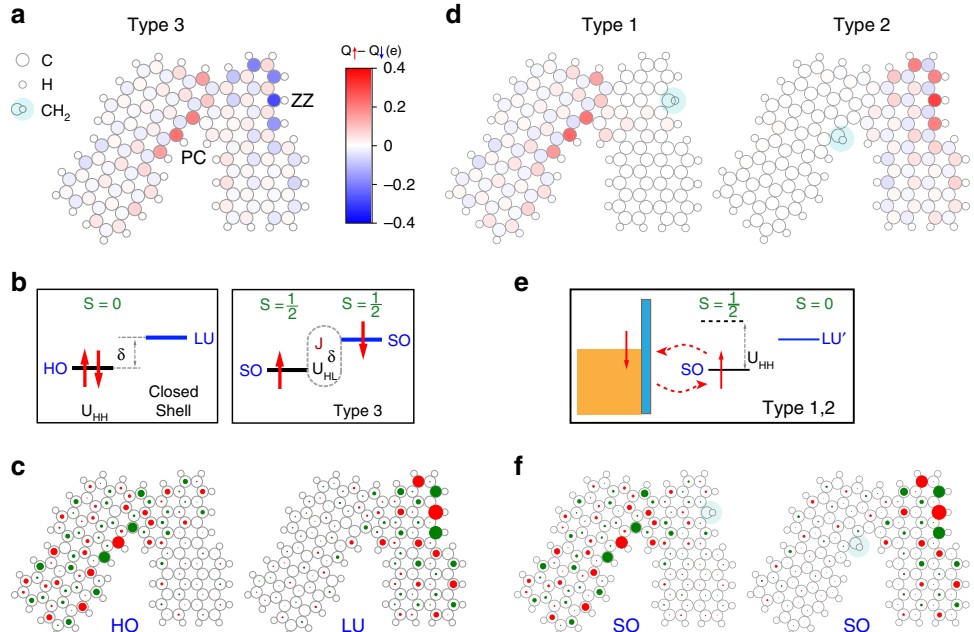

**Fig. 3** Calculated electronic states and spin polarization. **a** Spin polarization obtained from DFT simulations in a Mulliken population analysis. The standard junction (all peripheral carbons bonded to H) shows spontaneous spin localization in both PC and ZZ regions, revealing the apparition of radical states. **b** Schema of the spontaneous spin polarization when one of the two electrons in the HO level gets promoted to the LU level to form two separated, exchange coupled spin-1/2 systems (Type 3 junction). This process is energetically favored when the reduction in Coulomb energy $U_{HH} - U_{HL}$ plus exchange energy $J$ exceeds the level separation $\delta$, i.e., $\delta + U_{HL} - J < U_{HH}$. **c** Single-particle TB wave functions (HO/LU) for Type 3 junction. **d** Same as in **a** but now adding a H atom to an external carbon in either the ZZ (Type 1) or PC (Type 2). The passivation with H removes the corresponding radical state and, hence, its spin-polarization. **e** Sketch of the spin-1/2 Kondo state generated with a single radical (Type 1 and 2 junctions). **f** Single-particle TB wave functions (SO) for Type 1 and Type 2 junctions. Red-green colors represent the positive-negative phase

The observation of spin localization in only one of the two radical regions in Type 1 and 2 junctions implies that one of the two edge magnetic moments has vanished. Foreign atoms bonding to a SO $p_z$ orbital remove the local spin and suppress the magnetic signal at this site. Metal atoms can bind to C-sites, but the interaction is too weak to bind to $\pi$-radicals over a metal substrate[35]. Instead, H-passivation of radical sites is a highly probable process occurring on the surface due to the large amount of hydrogen available during the reaction[36]. DFT simulations show that attaching an extra H atom into an edge carbon in either the ZZ or PC sites leads to its $sp^3$ hybridization

and the removal of a $p_z$ orbital from the aromatic backbone. This completely quenches the magnetic moment of the passivated region (Fig. 3d), and leaves the junction with a single electron localized at the opposite radical site (Fig. 3e and Supplementary Fig. 7). According to this, a Type 1 junction shows Kondo at the PC site because it has a H atom bonded to the ZZ site that quenches that magnetic moment, and opposite for Type 2. The computed wave function amplitude distributions for the two energetically most favorable adsorption sites (Fig. 3f) are also in excellent agreement with the extension of the Kondo resonance mapped in Fig. 2a, b.

**Manipulation of the spin state of the nanostructures.** The presence of extra H atoms in Type 1 and 2 junctions was confirmed by electron induced H-atom removal experiments. Figure 4a shows a structure formed by three chGNRs connected via Type 1 and 2 junctions. Accordingly, their d$I$/d$V$ spectra (black curves in Fig. 4b, c) show a Kondo resonance at the PC$_1$ and ZZ$_2$ regions. We placed the STM tip on top of the opposite sites, ZZ$_1$ and PC$_2$, and raised the positive sample bias well above 1 V. A step-wise decrease of the tunneling current indicated the removal of the extra H atom (inset in Fig. 4b). The resulting junction appeared with double bright regions in low-bias images (Fig. 4d), and the PC$_1$ and ZZ$_2$ spectra turned into d$I$/d$V$ steps characteristic of Type 3 junctions (blue curves in Fig. 4b, c). Thus, the removal of H atoms activated the magnetic moment of the initially unpolarized ZZ$_1$ and PC$_2$ sites, converting Type 1 and 2 junctions into Type 3, and switching the total spin of the junction from spin to zero.

**Contacting the junctions with the STM tip.** The magnetic state of the junction was also changed by creating a contact between the STM tip apex and a radical site. $\pi$-radicals show some weak reactivity to bond metallic atoms, that allows their manipulation with an STM tip[37]. In the experiments shown in Fig. 5a, the STM tip was approached to the ZZ sites of a Type 3 junction. A step in the conductance-distance plot (Fig. 5b) indicated the formation of a contact. The created tip-chGNR contact could be stretched up to 3 Å before breaking (retraction step in Fig. 5b), signaling that a chemical bond was formed.

A reference d$I$/d$V$ spectrum recorded before the bond formation (black point in Fig. 5c) shows the split-peak feature of Type 3 junctions (black spectrum in Fig. 5c). After the bond formation (blue and red points in Fig. 5b), the spectra changed to show Kondo resonances (blue and red spectra in Fig. 5c), persisting during contact retraction until the bond-breaking step,

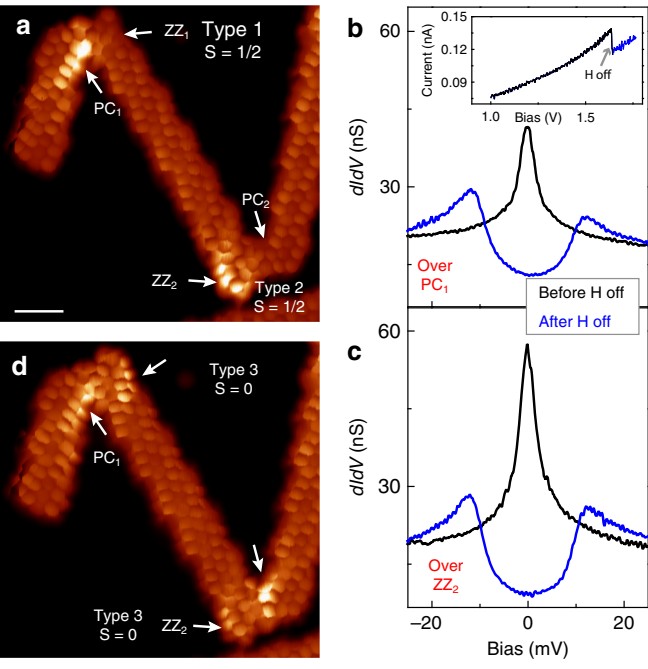

**Fig. 4** Spin manipulation by electron-induced removal of extra H-atoms. **a** Constant-height current image of two junctions with extra H atoms ($V =$ 8 mV) (scale bar 1 nm). **b**, **c** d$I$/d$V$ spectra taken over PC$_1$ and ZZ$_2$ regions (indicated in **a**) before (black) and after (blue) the dehydrogenation processes. Inset in **b** shows the current during the process of dehydrogenation. **d** Image with same conditions as in **a** after the electron-induced removal of the extra H-atoms. The dehydrogenation processes were done over the ZZ$_1$ and PC$_2$ sites

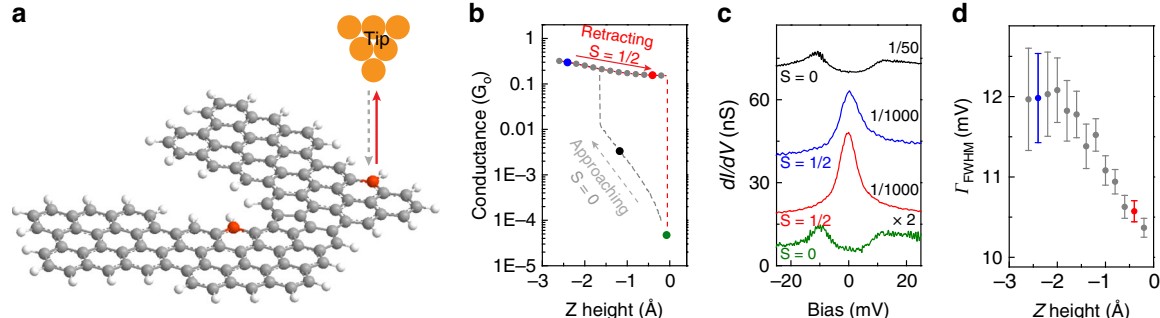

**Fig. 5** Kondo effect from the spin embedded in a lifted chGNR junction. **a** Schematics of the process where the tip of the STM is first approached to the ZZ site of a Type 3 junction (gray dashed arrow) and then retracted to lift the junction away from the substrate (red arrow), resulting in a suspended junction between tip and substrate. **b** Simultaneously recorded conductance curve ($V = −50$ mV) during the approach, jump to contact and lift processes. **c** d$I$/d$V$ spectra recorded at the specific heights indicated with colored points on the curves in **b**. **d** Full widths at half maximum (FWHM) of spectra acquired in the retraction process (points in **c**), extracted from a fit using the Frota function[49]

where double-peak features are recovered (green spectrum in Fig. 5c). The formation of a tip–chGNR bond thus removed the spin of the ZZ site, and the transport spectra reflect the Kondo effect due to the remaining spin embedded in the junction. If the STM tip contacts instead the ZZ radical site of a Type 2 junction (shown in Supplementary Note 5) the initial Kondo resonance disappears from the spectra, signaling the complete demagnetization of the junction. The width of the Kondo resonance in the contacted junctions (blue and red plots in Fig. 5c) is significantly larger than in Type 1 and 2 cases, probably because it incorporates scattering with tip states[38,39], and monotonously narrows as the contact is pulled apart (Fig. 5d). The survival of the Kondo effect in the contacted Type 3 junctions is a remarkable outcome of our experiments, which demonstrate the electrical addressability of localized magnetic moments in graphene nanostructure devices.

## Discussion

Open shell configurations of extended $\pi$ systems can be stabilized on-top of insulating layers[35,40]. The results presented here prove that the intrinsic open-shell character of a graphene nanostructure can survive on the surface of a metal. This is remarkable because it proves that key electron–electron correlations needed for the stabilization of magnetic ground states persist on the metal, in spite of the ubiquitous charge screening by the underlying substrate[41]. In addition, the adsorption on the Au(111) substrate has the general trend of hole-doping GNRs[42,43], which in some systems caused depopulation of the mid-gap states[37]. The band structure of the (3,1) chGNR and, in particular, the proximity of the VB to $E_F$, is a crucial aspect to stabilize the electron population of the GNR junction on the surface and hence, for the survival of their magnetic ground state.

## Methods

**Sample preparation and experimental details**. The experiments were performed on two different STMs operating in ultra-high vacuum. A commercial JT STM (from specs) operated at 1.2 K with a magnetic field up to 3 T was used to measure the temperature and magnetic field dependence of the Kondo resonance, while other experiments were done with a home made STM operating at 5 K. Both setups allow in situ sample preparation and transfer into the STM. The Au(111) substrate was cleaned in UHV by repeated cycles of $Ne^+$ ion sputtering and subsequent annealing to 730 K. The molecular precursor (2,2′-dibromo-9,9′-bianthracene) was sublimated at 170 °C from a Knudsen cell onto the clean Au(111) substrate kept at room temperature. Then the sample was first annealed at 200 °C for 15 min in order to induce the polymerization of the molecular precursors by Ullmann coupling, then the sample was annealed at 250 °C for 5 min to trigger the cyclodehydrogenation to form chiral graphene nanoribbons (chGNRs). A last step annealing at 350 °C for 1 min created nanostructure junctions. A tungsten tip functionalized with a CO molecule was used for high-resolution images. All the images in the manuscript were acquired in constant height mode, at very small voltages, and junction resistances of typically 20 MΩ. The d$I$/d$V$ signal was recorded using a lock-in amplifier with a bias modulation of $V_{rms} = 0.1$ mV (Fig. 2d, e) and $V_{rms} = 0.4$ mV at 760 Hz.

**Simulations**. We performed calculations with the SIESTA implementation[44] of DFT. Exchange and correlation (XC) were included within either the local (spin) density approximation (LDA)[45] or the generalized gradient approximation (GGA)[46]. We used a 400 Ry cutoff for the real-space grid integrations and a double-zeta plus polarization (DZP) basis set generated with an 0.02 Ry energy shift for the cutoff radii. The molecules, represented with periodic unit cells, were separated by a vacuum of at least 10 Å in any direction. The electronic density was converged to a stringent criterion of $10^5$. The force tolerance was set to 0.002 eV/Å. To complement the DFT simulations described above we also performed simulations based on the MFH model, known to provide a good description for carbon $\pi$-electron systems[7,8,15,16,47,48].

## Data availability

The data that support the findings of this study are available from the corresponding author upon reasonable request.

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

## Acknowledgements

We thank Manuel Vilas-Varela for the synthesis of the chGNR molecular precursor. We are indebted to Carmen Rubio, Dimas G. de Oteyza, Nestor Merino, Nicolás Lorente, Aran García Lekue, and Daniel Sánchez Portal for fruitful discussions. We acknowledge financial support from Spanish AEI (MAT2016-78293-C6, FIS2017-83780-P, and the Maria de Maeztu Units of Excellence Program MDM-2016-0618), the Basque Government (Department of Education, Grant PI-2015-1-42), the EU project PAMS (610446), the Xunta de Galicia (Centro singular de investigación de Galicia accreditation 2016-2019, ED431G/09), and the European Regional Development Fund (ERDF).

## Author contributions

J.L. and J.I.P. devised the experiment. D.P. designed the chGNR molecular precursor. J.L. realized the experiments. M.C. and D.J.C. assisted J.L. in part of the measurements, S.S. and T.F. did the theoretical simulations. All the authors discussed the results. J.L., T.F., and J.I.P. wrote the manuscript.

## Additional information

**Competing interests:** The authors declare no competing interests.

