## [Peer Review File · Nature Communications]

Reviewers' comments:

Reviewer #1 (Remarks to the Author):

J. Li and coworkers claim to experimentally observe and manipulate unpaired electron spins embedded within atomically precise graphene nanoribbons (GNRs). Using known on-surface growth strategies, they create (3,1) chiral GNRs (previously reported) and observe the existence of spontaneously fused GNR termini, which can contain one or two unpaired electron states. The authors then explore possible magnetic phenomena using scanning tunneling spectroscopy (STS) (i.e., the Kondo effect and inelastic spin excitations).

Though magnetism is predicted to exist in a variety of graphene structures, experimental demonstration of all-carbon magnetism remains relatively scarce. Previous studies have shown evidence of the existence of unpaired electron spins embedded in GNR termini (S. Wang, et al., 2016), but no direct observation of Kondo resonances nor exchange interactions have been performed previously on GNR-supported single-electron spins. Therefore, a bona fide demonstration of magnetism in GNR-based states as claimed here would be of significant interest to the field. I will also add that the data presented here are quite extensive and very impressive. The novelty and quality of this work is definitely at the level of Nature Communications. That said, I still feel that some essential issues must be better addressed before publication in Nature Communications (see points below).

1. In the last paragraph on p.8, the authors make an argument for why the inelastic gap observed in STS on type 3 junctions should not be attributed to single-particle states nor Coulomb-split radicals due to the observed scaling of the gap with system size. While this argument makes sense, it brings up the question of where (or if) the authors did observe single-particle states in STS (i.e. the elastic peaks for the localized states). One would expect a low-lying elastic peak occupying each termini in addition to the observed inelastic gap. Where is it? The authors' argument here would be much more convincing if elastic peaks that could be attributed to single particle states were also shown. If they cannot be parsed from the inelastic features, it becomes more difficult to distinguish an exchange-mediated gap (as the author's claim) from a regular hybridization gap.

2. The magnetic phenomena explored in this manuscript arise from unpaired pi electrons that appear at GNR termini. Though similar GNR end states have been observed previously (M. Koch, et al., 2012), they appear as elastic peaks at small positive bias near 0 V and Kondo resonances are ostensibly absent. The question here is similar to the one above: where are the expected elastic peaks for type 1 and type 2 junctions?

3. All previous examples of GNR end states observed on Au(111) suggest that they should be fully emptied due to charge transfer with the underlying substrate. This would remove any magnetic phenomena associated with unpaired electron spins. The authors should address this point, and explain why they believe this is not happening in their system.

Reviewer #2 (Remarks to the Author):

The manuscript entitled "Single Spin Localization and Manipulation in Graphene Open-Shell Nanostructures" by Li et al. contains an in-depth investigation of the local electronic properties of junctions of graphene nanoribbons grown on Au(111) via the combination of scanning tunneling spectroscopy and simulations using density functional theory and mean-field Hubbard model calculations. The very interesting new result is the experimental observation of Kondo features on particular sites of the junction, which, with the help of the calculations, can be unambiguously identified as arising from the spin-polarization of a single electron localized at certain regions of the graphene edges at the junction. Moreover, the spins at two different locations of the junction are shown to experience an exchange interaction which can be quantified via the measurements and depends on the lengths of the nanoribbons on either side of the junction. Finally, the spin-polarization can be induced at will via desorption of H atoms using the tip of the scanning

tunneling microscope as a tool.

The experimental search for theoretically predicted spin-polarizations at the edges of graphene has been a topic of strong interest since many years. So far, experimental proofs have been challenging and there are no fully convincing reports. Therefore, the results of the present manuscript of Li et al. are definitely of high interest to the general solid state physics community. I am thus convinced that the manuscript is suitable for publication in Nature Communications once the minor improvements specified below have been made. The manuscript is very well written and the interpretations and conclusions are free from flaws. There are just a few minor points that need to be improved for a better representation. These are specified point by point below.

- 1) Fig.2e shows a splitting of the Kondo resonance at 2.8 T, which could be analyzed and compared to the spin 1/2 model. Does it fit?
- 2) For the spectra measured on the ZZ sites (Fig.2c) the peak (step) at positive bias has a larger intensity than that on the negative bias side. For the spectra measured on the PC sites the situation is inverted. This seems to be a systematic effect, which could somehow relate to the spin-polarization and exchange coupling between the two spins at ZZ and PC. Please discuss.
- 3) Some fonts in the figures and some insets are too small and should be enlarged. In particular, the numbers in the color bar of Fig.1b and the insets in the two dashed circles in Fig.3b are tiny.
- 4) The acronym "chGNR" should be already defined on the bottom of the first paragraph of page 2.
- 5) There is a label "S=0" in the middle left part of Fig.4b, which seems to be superfluous.
- 6) The order of the different parts of the supplementary is rather random and should be sorted corresponding to the order in which the different topics appear in the main manuscript text.
- 7) The assignment of "ZZ" and "PC" to "type 1" and "type 2" is exchanged at different positions of the supplementary with respect to the main manuscript, e.g. in the first paragraph of page 3, in Fig.S2a, and in the last paragraph of page 6.
- 8) The nanoribbon arms left and right of the junction are named "a" and "b" in the supplementary on page 18 and in Fig.S15, but "L" and "R" in the main manuscript. This should be made consistent.

Reviewer #3 (Remarks to the Author):

In this manuscript the authors report a combined experimental and theoretical study of the 'single spin localization and manipulation in graphene open-shell nanostructures by performing scanning tunnelling microscopy, density functional theory and mean field theory Hubbard calculations.

Although the STM images and dI/dV measurements appear to be carefully made, I think the experimental interpretation is highly speculative.

It is well established in the chemistry that the free radicals employ the presence of unpaired electrons. These materials are highly reactive to the environment and therefore the "free radical" has an infinitely small life time that is not enough to be measured. It is also well known in chemistry that indeed some of these free radicals are can be stabilized by attaching bulky groups around that part of the molecule and therefore suppress their reactivity since no atom can come close to those unpaired electrons. This type of radicals has a lifetime long enough to be measured and characterized.

However, very important to note is that this is not the case for the systems used by the authors. The flat molecular structures used in this study do not have these bulky groups and within an STM experiment the free radical is exposed and will react with the metallic Au(111) surface or the STM tip. The fact that the authors synthesized these molecular structures onto the Au surfaces by means of Ullmann-like C-C coupling and cyclodehydrogenations also proves that the Au(111) is rather reactive and the interaction with these graphene like nanostructures should be quite strong.

The paper contains no real and direct proof of a Kondo resonance, the shape of the measured spectra might have other origin that are well discussed in literature.

Furthermore, the calculations are made for ideal molecular structures in the gas phase that are far away from the real experimental situation when the molecules are adsorbed on metallic surface.

Additionally, it is already well demonstrated by several experimental and theoretical studies that the edge states of the graphene-like nanoribbons strongly interact with the underlying substrate. Therefore, the interesting magnetic properties predicted by means of theoretical simulations for these nanoribbons in gas phase do not exist anymore when the nanoribbons are adsorbed onto surfaces.

As a note for the authors, I am convinced that the removal of the H-atom by the STM tip leads to a formation of a different (stronger) bonding of the C atom with the substrate.

To conclude, I do not recommend the publication of this manuscript in Nature Communications.

Reviewer #1

Though magnetism is predicted to exist in a variety of graphene structures, experimental demonstration of all-carbon magnetism remains relatively scarce. Previous studies have shown evidence of the existence of unpaired electron spins embedded in GNR termini (S. Wang, et al., 2016), but no direct observation of Kondo resonances nor exchange interactions have been performed previously on GNR-supported single-electron spins. Therefore, a bona fide demonstration of magnetism in GNR-based states as claimed here would be of significant interest to the field. I will also add that the data presented here are quite extensive and very impressive. The novelty and quality of this work is definitely at the level of Nature Communications. That said, I still feel that some essential issues must be better addressed before publication in Nature Communications (see points below).

We thank the Reviewer for appreciating the significance of our work. Indeed, the observation of a Kondo resonance is a spin-sensitive proof of the existence of localized magnetic moments.

1. In the last paragraph on p.8, the authors make an argument for why the inelastic gap observed in STS on type 3 junctions should not be attributed to single-particle states nor Coulomb-split radicals due to the observed scaling of the gap with system size. While this argument makes sense, it brings up the question of where (or if) the authors did observe single-particle states in STS (i.e. the elastic peaks for the localized states). One would expect a low-lying elastic peak occupying each termini in addition to the observed inelastic gap. Where is it? The authors' argument here would be much more convincing if elastic peaks that could be attributed to single particle states were also shown. If they cannot be parsed from the inelastic features, it becomes more difficult to distinguish an exchange-mediated gap (as the author's claim) from a regular hybridization gap.

The Reviewer is correct: the spin-polarized single particle states and their corresponding hybridization gap are expected to contribute with some weight in the spectral density. However, the tunneling spectra, measured in a wider range, do not provide conclusive results in this respect. This is probably due to the following reasons:

First, we note that their spectral weight is smaller and usually broader than the zero energy features. Thus, they are not so evidently seen in the energy scale and with the measuring conditions required to measure the Kondo resonance. Exploring a wider energy range, we find that the “pristine ribbon” VB and CB onsets are close to E_F , thus making it difficult to clearly discriminate its origin from the chGNR band structure in the STS spectra.

We can however provide a hint of their presence by mapping the dI/dV signal in a wide range of biases for similar tunneling conditions. The figure below shows that the large dI/dV signal around zero (at -35 meV and +40 meV) at the radical sites of a Type 1 and a Type 3 junctions, derived from the Kondo state, slowly decreases as the bias is shifted away from zero bias (i.e. at -135 and 140 mV) and shines back again at -185 mV (coinciding with the onset of the VB of the nearby GNRs) and on the PC sites at 360 mV (right below the onset of the CB).

The single particle states and their hybridization gap, however, cannot be confused with the inelastic steps and their peaked features that we observe in the experiment. As the Reviewer comments, we provide an important argument in our paper, namely, the energy scaling with the size is opposite as expected for single particle states. The magnitude of the hybridization gap is linked to the energy scale of Coulomb correlations and single particle spectrum, and both should be larger for smaller nanostructures. Exchange between the two spins, however, should depend inversely on their degree of wavefunction localization, which certainly increases for smaller nanostructures.

It is also important to note that the reduced energy scale of these zero-bias features and their proximity to zero-bias is not compatible with single-particle states in a graphene system on a metal surface. Resonances with a HWHM of less than 5 meV and spaced by a couple of meVs would mean a negligible interaction with the surface. Coulomb correlations should be large if this were the case and move peaks far from zero energy due to Coulomb charging. We also note that the steps around E_F appear at symmetric position around E_F in all spectra, a certain proof of their inelastic origin, and the slope above the excitation energy is similar to the logarithmic tail of the Kondo resonance (as shown e.g. in Figs. 2f, 4c and 4d), suggesting that Kondo-like correlations remain above the inelastic onset (e.g. as shown in Ref. 30 of our ms.). Hence, the uncertainty in the spectral determination of the (spin-polarized) single electron states cannot harm the identification of the inelastic origin of step spectral features.

2. The magnetic phenomena explored in this manuscript arise from unpaired pi electrons that appear at GNR termini. Though similar GNR end states have been observed previously (M. Koch, et al., 2012), they appear as elastic peaks at small positive bias near 0 V and Kondo resonances are ostensibly absent. The question here is similar to the one above: where are the expected elastic peaks for type 1 and type 2 junctions?

This question was answered above. We believe that the single-occupied (SO) and SU states lie in the energy region where the tail of CB and VB avoid their clear resolution. However, this cannot add uncertainty on the Kondo-origin of the zero bias resonance.

Change: To address better the reviewer's point in the manuscript we have added a discussion along these lines in a new section of the SI.

3. All previous examples of GNR end states observed on Au(111) suggest that they should be fully emptied due to charge transfer with the underlying substrate. This would remove any magnetic phenomena associated with unpaired electron spins. The authors should address this point, and explain why they believe this is not happening in their system.

The Reviewer is correct in the general tendency of GNRs to remain p-doped on a Au(111) surface, and the observation of a magnetic fingerprint here is indeed a surprising fact and a confirmation that for this graphene nanostructure the unpaired electrons remain. We speculate that this can also occur for other graphene systems.

The previously published examples solely refer to end resonances observed in armchair GNRs. In infinite 7AGNR the band gap is large, and its electron affinity promotes a certain alignment with the surface's E_F , which results that the mid-gap states are depopulated.

The behavior of the 7AGNR cannot be generalized. The (3,1) chGNR has a different band structure (see our Fig. S7-S8) and display no edge or mid-gap states at finite length. The singly occupied states found here are of a different nature. The equilibrium between ribbon's and surface's Fermi levels is also maintained by the chGNR body itself, which is also p-doped (VB close to E_F). However, the PC and ZZ states are not mid-gap end states as for the 7AGNRs, but simply two states split off from the VB and localized at the junction. Their alignment with respect the VB onset is such that they do not become depopulated after the band alignment.

Change: Following the referee's suggestion, we included the following sentence in the manuscript: "These are split-off states from the VB of the (3,1)-chGNR, which lies close below E_F \cite{Merino2017}." and a detailed discussion at the end of the manuscript.

Reviewer #2 (Remarks to the Author):

The experimental search for theoretically predicted spin-polarizations at the edges of graphene has been a topic of strong interest since many years. So far, experimental proofs have been challenging and there are no fully convincing reports. Therefore, the results of the present manuscript of Li et al. are definitely of high interest to the general solid state physics community. I am thus convinced that the manuscript is suitable for publication in Nature Communications once the minor improvements specified below have been made. The manuscript is very well written and the interpretations and conclusions are free from flaws. There are just a few minor points that need to be improved for a better representation. These are specified point by point below.

We thank the Reviewer for his/her understanding of the value of our work and for the help to polish some details of our manuscript.

1) Fig.2e shows a splitting of the Kondo resonance at 2.8 T, which could be analyzed and compared to the spin 1/2 model. Does it fit?

We do not see the complete split, but a broadening and the evolution of a flat plateau at the cusp that suggest we are close to the split. According to Costi et al PRL 85, 1504 (2000) a spin $\frac{1}{2}$ would show splits (for T sufficiently below T_k) at a critical magnetic field of $B \approx 0.5 T_k$. From the T-dependent measurements of Fig. 2d, we obtain that $T_k \approx 6$ K. This suggests that the complete split should occur around 3 T, in agreement with our measurements.

Change: We included in the Supporting Information a new section explaining the B-field and T dependence of the Kondo resonance and its implications.

2) For the spectra measured on the ZZ sites (Fig.2c) the peak (step) at positive bias has a larger intensity than that on the negative bias side. For the spectra measured on the PC sites the situation is inverted. This seems to be a systematic effect, which could somehow relate to the spin-polarization and exchange coupling between the two spins at ZZ and PC. Please discuss.

We associate such spectral asymmetries in the inelastic spectra to the degree of particle-hole asymmetry of every spin in its corresponding ribbon. As it has been previously reported (e.g. in Ref. 33 of our ms), the peaked rise at the inelastic step and the logarithmic fall off for biases above are attributed to the onset of Kondo-like fluctuation of the spin once the anisotropy energy has been overcome by tunneling electrons (i.e. out of equilibrium). The fact that these are more pronounced for particle tunneling (in ZZ) or for hole tunneling (in PC) suggest the proximity of the corresponding Singly Unoccupied or Singly Occupied state to EF, respectively.

As we obtain from our Tight Binding simulations (Fig. S12 and S13), the two orbitals at ZZ and PC are aligned differently with respect the ribbon's bands, and they have different degree of localization. Hence, it is reasonable that their singly-occupied configuration also presents the SO and SU states at different alignment respect to EF, out of particle-hole symmetry, which explains this different asymmetry.

Although these asymmetries are generally found in all spectra, only in some of the ribbons appeared as pronounced as in Fig. 2c. For example, in Fig. 4 the asymmetry is less apparent. We speculate that the different GNR length or adsorption configuration slightly affects the band's alignment and, thus, the degree of particle-hole asymmetry.

This explanation was left out of the original ms and now has been included in the following sentence.

Change: These arguments have been incorporated in the paragraph in page 5, starting with "Junctions with two bright regions.....", and the modified text is there indicated in red.

3) *Some fonts in the figures and some insets are too small and should be enlarged. In particular, the numbers in the color bar of Fig.1b and the insets in the two dashed circles in Fig.3b are tiny.*

This two cases have been updated in the revised version. We will carefully look for more of these cases at the proof-checking stage, if our paper is accepted for publication. We changed the insets of Fig. 3b to simply indicate a CH₂ configuration of that site.

4) *The acronym “chGNR” should be already defined on the bottom of the first paragraph of page 2.*

We have defined chGNR at the point the Reviewer suggests.

5) *There is a label “S=0” in the middle left part of Fig.4b, which seems to be superfluous.*

This label has now been removed.

6) *The order of the different parts of the supplementary is rather random and should be sorted corresponding to the order in which the different topics appear in the main manuscript text.*

In the resubmitted version, we have accounted for this comment raised by the Reviewer in part.

The Supplementary Information includes experimental and chemical material, DFT results, and HMF results. These are called in the main manuscript as they are needed to support the arguments. If we strictly follow this sequence, there will be not much order in the SI. We rather maintain these three parts separated (experimental material, DFT, HMF) and reordered the sections in each as they are called in the main text.

7) *The assignment of “ZZ” and “PC” to “type 1” and “type 2” is exchanged at different positions of the supplementary with respect to the main manuscript, e.g. in the first paragraph of page 3, in Fig.S2a, and in the last paragraph of page 6.*

We revised this point, which was correct but, in fact, is confusing. H-bonding to ZZ atoms, results in passivation of the magnetic moment localized at this site, and only the spin in the opposite PC sites survives, thus showing Kondo on PC sites and becoming Type 1 junction. Opposite to type 2 junctions. We slightly rephrase this

Change: *In first paragraph of Page 3, “ZZ sites are more favorable to incorporate an extra hydrogen atom and get passivated, becoming Type 1 junctions (22% of the radicals). The PC sites had an extra atom only in 13% of the cases, thus appearing as Type 2 junctions.”*

In the main text, we also added the following sentence in page 9: “According to this, a Type 1 junction shows Kondo at the PC site because it has a H atom bonded to the ZZ site that quenches that magnetic moment, and opposite for Type 2.”

8) *The nanoribbon arms left and right of the junction are named “a” and “b” in the supplementary on page 18 and in Fig.S15, but “L” and “R” in the main manuscript. This should be made consistent.*

We made now consistent these labeling and thank the Reviewer for his/her help in such critical reading of our manuscript.

Reviewer #3 (Remarks to the Author):

In this manuscript the authors report a combined experimental and theoretical study of the 'single spin localization and manipulation in graphene open-shell nanostructures by performing scanning tunnelling microscopy, density functional theory and mean field theory Hubbard calculations.

Although the STM images and dI/dV measurements appear to be carefully made, I think the experimental interpretation is highly speculative.

It is well established in the chemistry that the free radicals employ the presence of unpaired electrons. These materials are highly reactive to the environment and therefore the "free radical" has an infinitely small life time that is not enough to be measured. It is also well known in chemistry that indeed some of these free radicals are can be stabilized by attaching bulky groups around that part of the molecule and therefore suppress their reactivity since no atom can come close to those unpaired electrons. This type of radicals has a lifetime long enough to be measured

However, very important to note is that this is not the case for the systems used by the authors. The flat molecular structures used in this study do not have these bulky groups and within an STM experiment the free radical is exposed and will react with the metallic Au(111) surface or the STM tip. The fact that the authors synthesized these molecular structures onto the Au surfaces by means of Ullmann-like C-C coupling and cyclodehydrogenations also proves that the Au(111) is rather reactive and the interaction with these graphene like nanostructures should be quite strong.

We agree with the Reviewer that free radicals are reactive to environment and, hence, access to them by spectroscopy methods is an important challenge. For example, most organic radicals can easily react with O₂, H₂O, reaction partners or solvents when generated in solution under ambient conditions. However, under the ultra-clean environment of our low temperature and ultra-high vacuum setup, these radical states are stable for long time-scales, allowing detailed measurements. There are multiple previous observations of free-radical states in molecules and charge-transfer complexes (e.g. in Refs. 26 and 27 of our ms., or in Ang.Ch. Int. Ed. 2018, 57, 3888-3908) detected by STM. Therefore, bulky groups are not needed to stabilize radicals under UHV conditions.

In addition, the single electron states we study here derive from p_z carbon orbitals and therefore are π-radicals. The Reviewer probably refers to σ-radicals, which indeed are less stable than π-radicals and react strongly with the environment. In fact, as the reviewer mentions, these σ-radicals are created in our system in the dehalogenation step of the Ullmann-like-coupling reaction on the surface. At high temperature, Br endgroups detach from the precursor and the corresponding σ-radicals quickly react one to another. We always observe them passivated by the formation of either a C-C or a C-H bond. These σ-radicals are not the radicals we investigate here. Therefore, the survival of radical states on surfaces (in UHV) in the molecular systems studied in this ms. should not derive in controversy.

We would like to point out here that the Au(111) surface is not a specially reactive surface and that graphene nanostructures are known to generally lie in a rather weak physisorption state, with simply some amount of charge transfer with the metal, but no real chemical bond. The catalytic role of Au(111) in the on-surface reaction steps is only enhanced by annealing, and by the tendency of 3D molecules to adsorb planar on top of the metal surface.

The paper contains no real and direct prove of a Kondo resonance, the shape of the measured spectra might have other origin that are well discussed in literature.

We disagree with the Reviewer in this respect: our paper contains a direct proof of the Kondo resonance we claim. Probably the Reviewer overlooked the results we present in Fig. 2d and 2e. They are indeed a true experimental demonstration of the Kondo origin of the zero-bias resonance. Since the detailed description is rather technical, this was only briefly mentioned in the figure's caption with reference to previous papers.

Change in the manuscript: To emphasize on the proofs that the zero-bias resonance is a magnetic fingerprint of a localized spin $\frac{1}{2}$ we added some new sentences in page 4-5, to make reference to the anomalous T-dependent and to the B-dependence of the resonance in the main text, which were before only mentioned in the Figure's caption. We also included a new section in the Supplementary Information where experimental details that proves the Kondo origin of the resonance are more clearly explained.

For the shake of clarity, we briefly describe it here:

First we measured the broadening of the resonance with temperature. It is important to note that the plotted FWHM in Fig. 2d is corrected by the thermal broadening of the tip's Fermi edge. Hence, a bare single-electron resonance should show a temperature-independent line-width (constant line in the plot of Fig. 2d). The rather constant linewidth at very low temperatures followed by a linear broadening with temperature is a well established behavior of Kondo resonances.

Second (Fig. 2e) we measured the effect of an applied magnetic field on the narrow resonance, at constant temperature (1K). The magnetic-field induced decrease of the peak's intensity and its broadening is a proof of the magnetic origin of the zero-bias resonance. This can only be associated to a Kondo resonance.

The Reviewer also suggests that the zero bias features could have another origin, but he/she does not explicitly mention which. We can, however, discard that the Kondo resonance is due to single particle states, in case he/she refers to this. First, because as stated above in our reply to reviewer #1, a single particle peak should show a conventional electronic broadening with temperature and not be sensitive to the applied magnetic field. But also, we note that a hypothetical single particle state with a ≤ 1 meV line width would mean an unrealistically negligible interaction with the surface (electronic states in physisorbed systems are tens of meVs wide in tunneling spectra in the narrowest case). Furthermore, under such conditions, if such single-particle resonance would lie right a zero bias (singly occupied), it should appear split in spectra due to Coulomb e-e blockade in the tunneling process, certainly larger than 1 meV – as observed for molecular systems on insulators. So there is no way that a bare electronic resonance can appear as such narrow resonance at zero bias.

Furthermore, the calculations are made for ideal molecular structures in the gas phase that are far away from the real experimental situation when the molecules are adsorbed on metallic surface.

The presented calculations are oriented on interrogating the system about the origin of the magnetization, but they do not pretend to prove the Kondo origin of the resonances or demonstrate the survival of the magnetization on the surface. The unique proof of spin-polarization is the Kondo nature of this resonance (truly demonstrated by its anomalous thermal and B-field dependence).

Additionally, it is already well demonstrated by several experimental and theoretical studies that the edge states of the graphene-like nanoribbons strongly interact with the underlying substrate. Therefore, the interesting magnetic properties predicted by means of theoretical simulations for these nanoribbons in gas phase do not exist anymore when the nanoribbons are adsorbed onto surfaces.

We do not aim here at contradicting previous results or predictions, but our experimental results demonstrate that for the nanostructures shown here their intrinsic spin-polarization predicted by theory survives on the surface. This proves that paramagnetism may survive on a metal surface. This is indeed the impacting message of our manuscript.

As a note for the authors, I am convinced that the removal of the H-atom by the STM tip leads to a formation of a different (stronger) bonding of the C atom with the substrate.

We appreciate the alternative explanation provided by the Reviewer, which we have seriously considered.

If we interpret correct the Reviewer's comment, he/she is convinced that we remove an H atom from a peripheral C-H site (instead of from a C-H₂ site, as we claim), thus leaving a radical C site to make a chemical bond with the metal surface that produces the bright regions and the zero bias resonance (o the steps in Type 3 juncitons). The tip-removal of H was a test-experiment done on just a few of the sites. Most of the radicals we observed, and presented here, appeared on the sample naturally right after preparation of the system at elevated temperatures and cool down to low temperature. Hence, the Reviewer's comment would imply that, in fact, σ -radicals survive the on-surface reaction by bonding to the substrate.

We believe that this is very unlikely for the following reasons:

* Typically, π -aromatic planar platforms lie physisorbed at more than 0.3 nm above Au(111). A chemical bond of a terminal C with the surface is energetically unfavorable because it should bring this C down to less than 0.2 nm, the typical distance of a C-metal chemical bond. Such deformation would also distort the graphene platform by 0.1 nm, and this would be clearly seen in constant height current images.

* An eventual resonance caused by such strong chemical bond C-Au should present a large linewidth due to the strong hybridization of carbon and Au states, rather than a 1 mV wide resonance.

* Such state should not react to magnetic fields. In such a strong hybridization regime the magnetic fingerprint reported in Fig. 2 is not expected.

To conclude, I do not recommend the publication of this manuscript in Nature Communications.

We hope that our explanations above, and the new section added to the Sup. Inf., help to convince the Reviewer about the truly significance of our claim and its proof.

Reviewers' comments:

Reviewer #1 (Remarks to the Author):

The authors have adequately addressed our inquiries in their response and in their edits to the manuscript. The authors' experimental data is of high quality, and convincingly demonstrates the existence of carbon-based magnetism in graphene nanoribbons. We strongly support publication of this work in Nature Communications.

Reviewer #2 (Remarks to the Author):

I read the first round referee reports of the other two referees #1 and #3, the authors rebuttal of all three referee reports as well as the revised manuscript and supplementary.

The criticisms of all three referees including that of myself have been convincingly addressed. In particular, given the further analysis of the magnetic field dependence of the resonance of Fig. 2d and 2e, which the authors show in the revised supplement's section 3, there is no doubt that the experimentally observed feature is due to a Kondo resonance. This, to my opinion, unambiguously proves the presence of individual magnetic moments localized in the graphene nanoribbons, which is the very intriguing main claim of the present manuscript. This also notably refutes the arguments of the most critical referee #3.

I thus strongly recommend, that the revised version of the manuscript should be published in its present form at Nature Communications.

Reviewer #3 (Remarks to the Author):

The authors simply failed to answer the questions that I raised in the first referee report. While I do not want to repeat myself, I think that their arguments are not convincing and contradict the basics of organic chemistry reactivity.

The π -organic radicals are reactive radicals also in UHV conditions -- they are reactive towards the substrate they lie on. Therefore, up to now there is no real proof in the literature regarding the edge states in graphene nanostructures. Many publications claim the presence of edge states in graphene nanoribbons, but these studies report "hints" and use highly speculative interpretations (naive, too simplistic models). As a consequence, up to now there is no publication providing a real, hard experimental proof.

Why the Au(111) substrate is reactive enough to split strong sigma bonds in order to form these large graphene-like structures and suddenly is not reactive towards π -organic radicals? In my opinion this contradicts the common sense and the simplistic view of the authors is simply "alchemy".

If the substrate is not important (means that it can be neglected) and the authors are right with their interpretation, they should adsorb these molecules on graphene, boron nitride or other inert surfaces and get the same results --- that will be an undoubtable & hard experimental proof. I note here that this could be a breakthrough in this field.

As I wrote in my previous report, the STM images and dI/dV measurements appear to be carefully made, I also think that the experiments might indicate the presence of a Kondo system, BUT there no proof that the edge states in graphene nanostructures are responsible for Kondo resonance.

A simple search of the literature unveils that even a simple π -conjugated organic molecule (and

not a reactive π -organic radical!) adsorbed on a substrate with low reactivity can pick up adatoms of the surface and show a Kondo resonance at the molecular site.

In the study presented by the authors, why the reactive π -organic radicals would not interact with Au adatoms and form a Kondo system?

Reviewer #1 (Remarks to the Author):

The authors have adequately addressed our inquiries in their response and in their edits to the manuscript. The authors' experimental data is of high quality, and convincingly demonstrates the existence of carbon-based magnetism in graphene nanoribbons. We strongly support publication of this work in Nature Communications.

We thank the Reviewer for his/her statement that this work convincingly demonstrates carbon-based magnetism and for supporting the publication of our manuscript.

Reviewer #2 (Remarks to the Author):

I read the first round referee reports of the other two referees #1 and #3, the authors rebuttal of all three referee reports as well as the revised manuscript and supplementary.

The criticisms of all three referees including that of myself have been convincingly addressed. In particular, given the further analysis of the magnetic field dependence of the resonance of Fig. 2d and 2e, which the authors show in the revised supplement's section 3, there is no doubt that the experimentally observed feature is due to a Kondo resonance. This, to my opinion, unambiguously proves the presence of individual magnetic moments localized in the graphene nanoribbons, which is the very intriguing main claim of the present manuscript. This also notably refutes the arguments of the most critical referee #3.

I thus strongly recommend, that the revised version of the manuscript should be published in its present form at Nature Communications.

We thank the reviewer for his/her positive evaluation of our work. He/she acknowledges that the intriguing claim of the existence of magnetic moments in graphene sites is here unambiguously proved.

Reviewer #3 (Remarks to the Author):

The authors simply failed to answer the questions that I raised in the first referee report. While I do not want to repeat myself, I think that their arguments are not convincing and contradict the basics of organic chemistry reactivity.

We thank Reviewer 3 for the time dedicated to evaluating our work and regret that we could not fully convince him/her in our previous response. In this response, we have paid special attention to this last main concern, regarding the eventual presence of Au adatoms bound to the graphene radicals.

The π -organic radicals are reactive radicals also in UHV conditions -- they are reactive towards the substrate they lie on. Therefore, up to now there is no real proof in the literature regarding the edge states in graphene nanostructures. Many publications claim the presence of edge states in graphene nanoribbons, but these studies report "hints"

and use highly speculative interpretations (naive, too simplistic models). As a consequence, up to now there is no publication providing a real, hard experimental proof.

Why the Au(111) substrate is reactive enough to split strong sigma bonds in order to form these large graphene-like structures and suddenly is not reactive towards π -organic radicals? In my opinion this contradicts the common sense and the simplistic view of the authors is simply “alchemy”.

The Au(111) surface is a rather inert surface and that graphene nanostructures are known to generally lie in a rather weak physisorption state, with simply some amount of charge transfer with the metal, but no real chemical bond. The reactivity of Au(111) for breaking sigma bond during on-surface reaction steps is only present at very high temperatures, where molecular dynamics are thermally excited and, is also enhanced by the tendency of 3D molecules to adsorb planar on top of the metal surface.

If the substrate is not important (means that it can be neglected) and the authors are right with their interpretation, they should adsorb these molecules on graphene, boron nitride or other inert surfaces and get the same results --- that will be an undoubtable & hard experimental proof. I note here that this could be a breakthrough in this field.

We agree with the referee in the breakthrough that that proposed experiment would represent. However, as he/she mentions above, there is to date no direct spin-sensitive proof that demonstrates edge magnetism in graphene nanostructures. Hence, the bare demonstration of spin localization in these graphene nanoribbons on Au(111) is by itself a breakthrough. It also demonstrates that graphene magnetism may survive even over a metal surface.

As I wrote in my previous report, the STM images and dI/dV measurements appear to be carefully made, I also think that the experiments might indicate the presence of a Kondo system, BUT there no proof that the edge states in graphene nanostructures are responsible for Kondo resonance.

A simply search of the literature unveils that even a simply π -conjugated organic molecule (and not a reactive π -organic radical!) adsorbed on a substrate with low reactivity can pick up adatoms of the surface and show a Kondo resonance at the molecular site.

In the study presented by the authors, why the reactive π -organic radicals would not interact with Au adatoms and form a Kondo system?

To answer this last question: “*why the reactive π -organic radicals would not interact with Au adatoms and form a Kondo system?*” we list the following points to demonstrate that this scenario cannot occur, is not observed, and would not explain our results:

1- Weak chemical reactivity of π -radicals: The reactivity of π -radicals on metals under UHV conditions is weak and strongly depends on the structure of the molecule/nanostructure. Very recently, Pavlicek et al. have generated triangulene on different surfaces under UHV conditions (Nature Nanotechnology volume12, pages 308–311 (2017), DOI: 10.1038/NNANO.2016.305). Triangulene is an open-shell molecule with π -diradical character, but still they were able to characterize it on Cu(111), a slightly more reactive surface than Au(111). The authors explicitly comment in the paper:

“The AFM data unambiguously demonstrate stable adsorption without any signatures of chemical bonding to the supporting Cu surface. Instead, the brighter appearance (less attractive) of the peripheral carbons suggests a slightly bent adsorption with the outer carbons further away from the surface, as observed previously for pentacene on Cu(111) (ref. 18). This observation is in strong contrast to previously investigated diradicals that form strong covalent bonds when adsorbed on copper under the same conditions (ref. 14,15), and can be rationalized by the fact that these comprise σ radicals, whereas tri-angulene features π radicals.”

This situation described above is similar to our open-shell graphene nanostructure. In fact, in our case the pi-radicals are more delocalized within a larger aromatic system, therefore it can be expected less reactivity towards the substrate. The Au(111) substrate is also more inert than Cu(111). Hence, in terms of chemical reactivity, our results are more than consistent with previous observations regarding the absence of bonding to adatoms extracted from the bare metal underneath.

2- A gold atom bonded to the pi-radical would quench its magnetic moment and, hence, would not show the Kondo effect. The gold 6s1 orbital would hybridize with the pi-radical and quench its magnetic moment, similar as we found that H-atoms do. So, we exclude this. On the contrary, as the Reviewer also mentions, an Au atom might even bond to “... a simply pi-conjugated organic molecule”, i.e. a non-radical C atom, and might induce in this case a radical, but this could happen similarly at any other carbon site of the graphene nanoribbons, rather than at this peculiar two sites that we observe. So, we experimentally exclude this scenario too.

3- We do not have any evidence of an Au adatom bonded to the graphene junctions. The high-resolution images would show Au adatoms if these were present. We find atoms neither sideways, nor underneath (an atom underneath would considerably lift the junction sides, which is not the case). Furthermore, our experimental observations are in striking agreement with theory simulations. The experimental spatial distribution of the Kondo amplitude fits with the distribution of the spin-polarization, and the excitation energy of type-3 junctions fit with the simulated dependence on the junction size. This would not be the case if a single Au adatom would lie at a specific site.

4- pi-radicals indeed may interact with Au adatoms. We know this from our manipulations experiments, where the STM tip is approached to form a bond to the radical sites (Fig. 5 and Fig. S5). This bond is weak, and quickly breaks after the tip is retracted a few Angstroms. On the contrary, such manipulation experiments would not be possible if a carbon edge sites were fully saturated with four bonds to the two C neighbors, a H atom and the Au adatom. The tip apex would not be able of forming an additional bond here. Additionally, we never observed any Au adatom left near the junctions after these tip manipulations experiments. So, we can exclude that the bright sites are due to Au adatoms bond to the edge C atoms.

Our arguments 3 and 4 clearly excludes the Reviewer scenario on the basis of our experimental results. Furthermore, argument 2 excludes that the scenario proposed by the Referee would give a Kondo resonance at all, which is our main fingerprint. Finally, argument 1 excludes that the spontaneous bond of pi-radical to adatoms extracted from the surface can occur from previous observation of highly more reactive situations.

These points could make our manuscript more complete and, hence, we included them in the following revision:

- In the subsection entitled “Contacting the junctions with the STM tip.” We modified the first paragraph to mention that *“pi-radicals show some weak reactivity to bond metallic atoms, that allows their manipulation with an STM tip \cite{Koch2012}.”* We further added *“Such contacts only succeeded over bright PC and ZZ sites, revealing their pi-radical character, while over passivated sites failed.”* After this paragraph, it is implicitly clear that if an Au adatom were responsible of such bright areas this would not explain the reactivity we observe to contact the STM tip, as we mention in point 4.
- In subsection “Theory simulations to uncover the origin of spin polarization” we included a sentence stating: *“The spin distribution along the edge sites reproduces the distribution of dl/dV signal measured in Type 3 junctions. This supports that this signal is an intrinsic effect of junction edge sites, rather than caused by defects or other atomic species.”* This sentence presents some of the arguments of point 3 above.
- In the last paragraph of the “Theory simulations...” section, we included the following sentence: *“... Foreign atoms bonding to a singly occupied p_z orbital remove the local spin and suppress the magnetic signal at this site. Metal atoms can bind to C-sites, but the interaction is too weak to bind to pi-radicals over a metal substrate \cite{Pavlicek2017c}. Instead,... ”* With this sentence we clearly state that i) as in the work by Pavlicek et al, pi-radicals do not bind to metals, ii) that if this were the case, the metal would passivate the radical, thus combining points 1 and 2 from above.